In silico engineering of Pseudomonas metabolism reveals new biomarkers for increased biosurfactant production

Occhipinti Annalisa 1
Eyassu Filmon 1
Rahman Thahira J. 2
http://orcid.org/0000-0001-7416-4372 Rahman Pattanathu K. S. M. 2 3
Angione Claudio 1 c.angione@tees.ac.uk
1 Department of Computer Science and Information Systems, Teesside University , Middlesbrough , UK
2 Technology Futures Institute, School of Science, Engineering and Design, Teesside University , Middlesbrough , UK
3 Institute of Biological and Biomedical Sciences, School of Biological Sciences, University of Portsmouth , Portsmouth , UK
Scheibe Timothy
Electronic publication date: 2018 Dec 17
Publication date: 2018
Volume: 6
Electronic Location ID: e6046
Received 2018 Apr 2; Accepted 2018 Oct 30
Copyright: © 2018 Occhipinti et al.
Copyright year: 2018
Copyright holder: Occhipinti et al.
License: This is an open access article distributed under the terms of the Creative Commons Attribution License, which permits unrestricted use, distribution, reproduction and adaptation in any medium and for any purpose provided that it is properly attributed. For attribution, the original author(s), title, publication source (PeerJ) and either DOI or URL of the article must be cited.
License URL: https://creativecommons.org/licenses/by/4.0/

Keywords: Pseudomonas, Rhamnolipids, Biosurfactants, Flux balance analysis, Metabolic engineering, Pseudomonas putida, Multi-omics, Genome-scale metabolic model, Regression, Machine learning

Funding: BBSRC, grants CBMNet-BIV-D0097 and CBMNet-PoC-D0156 with in-kind time contribution from TeeGene Biotech Ltd Teesside University Grand Challenge Funding award This work is supported by BBSRC, grants CBMNet-BIV-D0097 and CBMNet-PoC-D0156 with in-kind time contribution from TeeGene Biotech Ltd. The authors also received a Teesside University Grand Challenge Funding award. The funders had no role in study design, data collection and analysis, decision to publish, or preparation of the manuscript.

==============================
Background

Rhamnolipids, biosurfactants with a wide range of biomedical applications, are amphiphilic molecules produced on the surfaces of or excreted extracellularly by bacteria including Pseudomonas aeruginosa. However, Pseudomonas putida is a non-pathogenic model organism with greater metabolic versatility and potential for industrial applications.

Methods

We investigate in silico the metabolic capabilities of P. putida for rhamnolipids biosynthesis using statistical, metabolic and synthetic engineering approaches after introducing key genes (RhlA and RhlB) from P. aeruginosa into a genome-scale model of P. putida. This pipeline combines machine learning methods with multi-omic modelling, and drives the engineered P. putida model toward an optimal production and export of rhamnolipids out of the membrane.

Results

We identify a substantial increase in synthesis of rhamnolipids by the engineered model compared to the control model. We apply statistical and machine learning techniques on the metabolic reaction rates to identify distinct features on the structure of the variables and individual components driving the variation of growth and rhamnolipids production. We finally provide a computational framework for integrating multi-omics data and identifying latent pathways and genes for the production of rhamnolipids in P. putida.

Conclusions

We anticipate that our results will provide a versatile methodology for integrating multi-omics data for topological and functional analysis of P. putida toward maximization of biosurfactant production.

Introduction

The growing demand for rhamnolipids production owes to its wide range of industrial and biomedical applications, including pharmaceuticals, cosmetics and detergents (Randhawa & Rahman, 2014). The rhamnolipids composed of glycosyl head group (i.e., rhamnose moiety) and fatty acid (FA) tail, well-characterized bacterial biosurfactants, are mainly produced by Pseudomonas aeruginosa (Rahman et al., 2002; Abdel-Mawgoud, Lépine & Déziel, 2014; Randhawa & Rahman, 2014). P. aeruginosa, a gram-negative opportunistic bacterial pathogen, is widely studied for the biosynthesis of rhamnolipids. The production of these biosurfactants relies on two precursors: L-rhamnose and R-3-hydroxy FA, an intermediate of the FA degradation pathway. The rhamnosyltransferase A (RhlA encoded by rhlA) dimerizes R-3-hydroxy FAs to form R-3-((R-3-hydroxyalkanoyl)oxy)alkanoic acids (HAA) (Déziel et al., 2003; Zhu & Rock, 2008; Abdel-Mawgoud, Lépine & Déziel, 2014); subsequently, the rhamnosyltransferase RhlB catalyzes the addition of the first rhamnose moiety, forming mono-rhamnolipids (Rahim et al., 2001; Abdel-Mawgoud, Lépine & Déziel, 2014). By contrast, Pseudomonas putida is a gram-negative, soil-dwelling, non-pathogenic bacterium and represents a model organism with versatile metabolism with valuable industrial applications (Wittgens et al., 2011; Tiso et al., 2016). Although it is an evolutionary close relative of P. aeruginosa, its simplified genetics, the lack of complex regulatory networks found in P. aeruginosa and the presence of pathways required for the synthesis of rhamnolipid precursors made P. putida the ideal bacterium of choice to conduct this study.

The applications of bacterial surfactants are diverse and rapidly growing in demand. One of the reasons rhamnolipids have become such an attractive area for biochemical research is the scope of their applications. Rhamnolipids could replace petrochemical derived surfactants used in many cleaning products detergents (Randhawa & Rahman, 2014). Rhamnolipids have also been shown to be a valuable resource in the agricultural industry, providing pest resistance in various plants, for example, stimulating the expression of important defense genes in tobacco plants and protecting monocotyledonous plants against harmful biotrophic fungi (Mulligan, 2005). Additionally, it has been shown that they are able to improve nutrient adsorption in plant roots (Sachdev & Cameotra, 2013). The emulsifying properties of rhamnolipids make them an ideal tool for the bioremediation of oil spills. Rhamnolipids are extremely effective in aiding removing oil from contaminated soil and facilitating its breakdown and dispersal in aqueous environments (Chen et al., 2013; Kosaric, 2001). Due to their low toxicity, high bio-degradability and environmental compatibility, rhamnolipids are used efficiently in microbial enhanced oil recovery and are invaluable tool in bioremediation efforts (Amani, 2015).

Perhaps one of the most interesting applications of rhamnolipids is within cosmetic and pharmaceutical industries. Rhamnolipids show potential to be used in a range of cosmetics such as moisturizers, shampoo, lubricants and anti-wrinkle creams (Randhawa & Rahman, 2014). Research has shown rhamnolipids to have antimicrobial activities against a host of human pathogens such as Gram-negative bacteria (Salmonella typhimurium, Escherichia coli, Enterobacter aerogenes, Serratia marcescens and Klebsiella pneumoniae), Gram-positive bacteria (Listeria monocytogenes, Staphylococcus aureus, S. epidermidis, Bacillus cereus and B. subtilis) and fungi (Phytophthora infestans, Phytophthora capsici, Botrytis cinerea, Fusarium graminearum, Mucor spp., Cercospora kikuchii, Cladosporium cucumerinum, Colletotrichum orbiculare, Cylindrocarpon destructans and Magnaporthe grisea) (Rodrigues et al., 2006; Magalhaes & Nitschke, 2013). In addition to this, patents have been obtained for the use of rhamnolipids to treat organ transplants rejection, atherosclerosis, depression, schizophrenia, burn shock, wound healing (Piljac & Piljac, 2007). The use of rhamnolipids in these industries may make their commercialization economically viable. The higher profits that could be made per gram of rhamnolipids produced when compared with other industries such as cleaning products or bioremediation mean that the high costs of production and low yields would be less significant. This would come with a whole new set of challenges, as rhamnolipids being produced for pharmaceuticals would need to be of an extremely high purity.

Several factors affect the quality and quantity of rhamnolipids produced, the most important being the carbon source and the nutrient medium. Carbon sources such as glycerol, glucose, sucrose, mannitol, aliphatic and aromatic hydrocarbons have been successfully used for rhamnolipid production by Pseudomonas spp. (Silva et al., 2010; Puskarova et al., 2013). Although the use of low-cost materials is usually considered to solve the cost problem, the selection of a substrate compatible with cell growth is very important.

The aim of this study is to investigate the metabolic capabilities of P. putida for rhamnolipids biosynthesis using multi-omics modelling, statistical, metabolic and biosynthetic engineering approaches. We explore the techniques used by Wittgens et al. (2011) and Tiso et al. (2016) by introducing the RhlA and RhlB genes from P. aeruginosa to reconstruct an engineered genome-scale model of P. putida. Genome-scale constraint-based models have been constructed and applied extensively to a range of problems: genome annotation (Ganter, Kaltenbach & Stelling, 2014), comparative analyses (Oberhardt et al., 2011; Monk et al., 2013; Bartell et al., 2014; Babaei, Ghasemi-Kahrizsangi & Marashi, 2014; Van Heck et al., 2016; Koehorst et al., 2016), analyses of omics data (Colijn et al., 2009; Chandrasekaran & Price, 2010; Zur, Ruppin & Shlomi, 2010; Vijayakumar et al., 2017), disease (and cancer) characterization (Eyassu & Angione, 2017; Aurich, Fleming & Thiele, 2017; Angione, 2018), drug discovery (Plata et al., 2010) and metabolic engineering (Puchałka et al., 2008; McAnulty et al., 2012; Kim et al., 2015).

We simulate single objectives using linear programming, focusing on biomass and rhamnolipids production. We further investigate the flux distributions using statistical and machine learning techniques to elucidate the role of the individual reactions and pathways in determining the predicted phenotype. Our predictions can be used in synthetic biology to suggest optimal steps for engineering microorganisms and for analyzing complex omic networks. We finally present a methodological framework to integrate and analyze gene expression data in the contest of the metabolic model, in order to closely investigate the pathways and reactions involved in the production of rhamnolipids. To the best of our knowledge, this is the first study that uses multi-omics in silico modelling of P. putida for optimizing rhamnolipids synthesis.

Methods

Reconstruction of the engineered constraint-based genome-scale metabolic model

To reconstruct a metabolically engineered model of P. putida for rhamnolipids production, following Wittgens et al. (2011) and Tiso et al. (2016), we introduced two pathways for rhamnolipids biosynthesis by collating the full list of reactions catalyzed by RhlA and RhlB to an existing genome-scale model of P. putida, iJP962 (Oberhardt et al., 2011). The RhlA and RhlB genes produce rhamnolipids by three sequential reactions (Fig. 1A). RhlA is involved in the synthesis of HAA (Déziel et al., 2003), and is loosely bound to the inner membrane (Rahim et al., 2001). The next reaction is catalyzed by the membrane-bound RhlB rhamnosyltransferase and uses dTDP-L-rhamnose and an HAA as precursors, yielding mono-rhamnolipids (Rahim et al., 2001). The RhlA and RhlB genes are clustered with rhlR and rhlI, which encode proteins involved in their transcriptional regulation through the quorum-sensing (QS) response and they are arranged as an operon. RhlI and LasI synthesize the QS autoinducer molecules butanoyl-homoserine-lactone (C4-HSL) and 3-oxo-dodecanoyl-homoserine-lactone (3-oxo-C12-HSL), respectively. When their concentration reaches a threshold, they bind to the regulator proteins and induce the expression of the Rhl-genes (Wittgens et al., 2017). QS response regulates the production of rhamnolipids (Van Delden & Iglewski, 1998), as well as hundreds of additional genes (Hentzer et al., 2003; Schuster et al., 2003; Wagner et al., 2003).

Figure 1 Biosynthesis of rhamnolipids from the metabolic engineered model of Pseudomonas putida.

(A) Rhamnolipids biosynthesis pathway. The figure depicts central carbon metabolism, glycolysis and the tricarboxylic acid (TCA) cycle, and two rhamnolipids precursor pathways: fatty acid (FA) degradation pathway and the rhamnose pathway. Two genes (rhlA and rhlB) and their corresponding reactions from P. aeruginosa were incorporated into the P. putida model. Myristic acid is metabolized through the FA degradation pathway, generating intermediates, where RhlA and RhlB sequentially generate rhamnolipids. On the other hand, RhlB synthesizes rhamnolipids through the rhamnose pathway at a higher flux rate (depicted by the line thickness) whereby the uptake of glucose, fructose, sucrose and glycerol was used as the main carbon sources. Dotted lines represent multistep reactions, while the line thickness represents the relative flux carried by the corresponding pathway. (B) Rate of biomass and rhamnolipids production from the P. putida model simulated under different carbon sources. Biomass and rhamnolipids production increase linearly with the rate of substrate uptake. Myristic acid (C-14) provided maximum biomass and rhamnolipids production compared to the other carbon sources. (C) Rhamnolipids production by the P. putida model from the rhamnose pathway and fatty acid (FA) pathway. Under all conditions, the rhamnose pathway generates maximum amount of rhamnolipid. (D) Optimization for biomass and rhamnolipids production by the P. putida model. Under each carbon source, high rate of rhamnolipids was synthesized through the rhamnose pathway. (E) Comparison of rhamnolipids production in the P. putida and P. aeruginosa. Under each carbon source, the rate of rhamnolipids production was higher in P. putida than P. aeruginosa.

The full set of known biochemical reactions for rhamnolipids biosynthesis were added from the P. aeruginosa model (Oberhardt et al., 2008) to the P. putida model iJP962. Where appropriate, stoichiometrically balanced reactions of the rhamnose pathway from KEGG (Kanehisa & Goto, 2000), MetaCyc (Caspi et al., 2016) and BRENDA (Schomburg et al., 2013) were added. Table 1 shows the reactions for rhamnolipids biosynthesis that were added to the P. putida model. Reactions RHLA, RHLB and RHLC represent the rhamnosyltransferase chain A, rhamnosyltransferase chain B and rhamnosyltransferase 2, respectively. Reactions 3H3H and PHAC are involved in the poly (3-hydroxyalkanoic acid) synthase (Oberhardt et al., 2008). The reaction flux across inner and outer membranes was carried out by transport reactions, which were modelled as reactions converting intracellular into extracellular compounds. A transport reaction was also added for the export of rhamnolipids across the cell membrane. For the full list of reactions for rhamnolipids synthesis, exchange and transport see Additional File 1.

Table 1 List of reactions for rhamnolipids biosynthesis added to the P. putida model.

Code	Reaction formula	Reversibility	
RHLA	(3R)-3-Hydroxydecanoyl-acyl-carrier protein + Coenzyme A ⇒ (S)-3-Hydroxydecanoyl-CoA + acyl carrier protein	Irreversible	
RHLB	3-hydroxydecanoyl-3-hydroxydecanoate + dTDP-4-dehydro-6-deoxy-L-mannose + H+ ⇒ dTDP + L-rhamnosyl-3-hydroxydecanoyl-3-hydroxydecanoate	Irreversible	
RHLC	dTDP-4-dehydro-6-deoxy-L-mannose + H+ + L-rhamnosyl-3-hydroxydecanoyl-3-hydroxydecanoate ⇒ dTDP + L-rhamnosyl-Lrhamnosyl-3-hydroxydecanoyl-3-hydroxydecanoate	Irreversible	
3H3H	2 beta-hydroxydecanoyl-beta-hydroxydecanoyl-S-CoA + H2O ⇒ 3-hydroxydecanoyl-3-hydroxydecanoate + Coenzyme A	Irreversible	
PHAC	(S)-3-Hydroxydecanoyl-CoA ⇒ 2 beta-hydroxydecanoyl-beta-hydroxydecanoyl-S-CoA + Coenzyme A	Irreversible	
Note:

Reactions RHLA, RHLB and RHLC represent the rhamnosyltransferase chain A, rhamnosyltransferase chain B and rhamnosyltransferase 2 respectively. Reactions 3H3H and PHAC are involved in the poly(3-hydroxyalkanoic acid) synthase.

The model was then manually curated to establish that the new reactions were fully integrated. This was achieved by evaluating metabolite specificity and metabolite charges accordingly, as well as reaction directionality to confirm that each reaction carried a flux. Gene-protein-reaction rules of the added reactions were also curated from literature (for the final metabolic model see Additional File 2). To run the model, a linear optimization for rhamnolipid production was then carried out, and flux balance analysis (FBA) (see the following subsections for a detailed description) was used to analyze the newly reconstructed engineered model.

Geometric flux balance analysis

Flux balance analysis is a widely used mathematical approach for modelling large-scale metabolic networks (Orth, Thiele & Palsson, 2010). Because FBA assumes the homeostasis of a system, it does not require knowledge of metabolite concentrations and enzyme kinetics. This differentiates FBA from other modelling techniques that require kinetic parameters, usually difficult to obtain. In FBA, the set of biochemical reactions is represented mathematically in the form of a stoichiometric matrix (S) with dimensions of m × n, where the m metabolites are represented a rows and the n reactions are represented as columns. The stoichiometric matrix is a numerical matrix of stoichiometric coefficients for each metabolite participating in a reaction. The stoichiometric coefficient for every metabolite consumed and produced in the system has a negative and positive coefficient respectively. A zero stoichiometric coefficient is given for every metabolite that does not take part in a given reaction. It is assumed that the system is at a pseudo-steady state S · v = 0 that holds for internal metabolites, i.e. those reactants and products of the chemical reactions constituting the model that cannot be imported or exported directly. The vector v represents the flux distribution of the n reactions. Exchange metabolites can be imported and exported from the system, so they do not satisfy the steady state assumption. This flux distribution v therefore represents a feasible flux of metabolites through the reaction network, where under the principle of mass conservation the total amounts of internal metabolite consumed and internal metabolite produced are the same, and the derivative of their concentration is therefore equal to zero.

Constraints such as directionality and capacity (based on enzyme activity, Gibbs free energy change and uptake rates from the literature) are placed on individual reactions by defining the upper (Vmax) and lower (Vmin) bounds on the range of values that the flux of each reaction can hold (Vmin ≤ v ≤ Vmax). These constraints define the space of allowable flux distributions at which every metabolite is consumed or produced by each reaction in the system. Despite these constraints, the system is still underdetermined (there are more unknowns than equations), and therefore infinite possible solutions exist. A flux distribution can be obtained by defining an objective function that is a scalar product of the vector of flux rates v, and a vector of weights c, measuring how each component in the network contributes to the production of a biologically desirable phenotype. The set of all possible solutions to the FBA problem is given by the equation and constraints: (1) max c⋅v, such that S⋅v=x˙ x˙i=0 if Mi∈internal metabolites x˙i∈ℝ if Mi∈exchange metabolites Vmin≤v≤Vmax

where in our case the vector c allows to select either the biomass or rhamnolipids as the objective function.

In our pipeline, we use the geometric flux balance approach to define a unique flux balance solution (Smallbone & Simeonidis, 2009). Geometric FBA is based on a geometric representation of a FBA problem. In particular, every FBA problem defines a polyhedron which can be naturally decomposed as the sum of a convex hull and a pointed cone; FBA solutions are to be found within the hull. Since the vertices of the hull and the rays of the cone are uniquely defined, the center of the solution hull (i.e., the final FBA solution) is uniquely defined. Using the geometric FBA algorithm allows us to choose a unique and well-defined flux from the space of all possible solutions. The solution provided also satisfies a number of additional constraints. Indeed, the model assumes that flux correlates with enzyme levels, which is equivalent to the cell minimizing the amount of enzyme required to satisfy this objective. Moreover, the algorithm removes any fluxes representing thermodynamically infeasible internal cycles and selects the solution required to satisfy the given objective from the remaining set of solutions. Hence, the chosen unique solution flux is in a sense “central” and can be considered unbiasedly representative of all possible FBA solutions.

Objective function and uptake rates for optimal rhamnolipids synthesis

Consistent with the reference model iJP962, we used the uptake of glucose at 10 millimoles per gram dry weight per hour (mmol/gDW/h) as a control growth condition. To determine the best carbon source for optimal rhamnolipids synthesis, we investigated alternative carbon sources separately: fructose, sucrose, glycerol, benzoate and myristic acid. We simulated the growth medium with a single carbon source by setting to 10 mmol/gDW/h the uptake of the carbon source under investigation and to zero the uptake of the other sources.

Our P. putida model was optimized to maximize the production and export of rhamnolipids. Hence, we used maximum rhamnolipids production as the objective function in our engineered model. Geometric FBA was used to calculate the optimal flux distribution that maximizes the objective function. Simulations were carried out in MATLAB (version R2018a) using the COBRA toolbox (Schellenberger et al., 2011), with the linear programming solver GLPK (the Matlab script is provided as Additional File 3).

Using gene expression data to build condition-specific metabolic models

Understanding how the transcriptomic alterations change the metabolic phenotype can provide an effective method for data interpretation and analysis (Vijayakumar et al., 2018; Stephens et al., 2015). To this end, we also used the P. putida metabolic model to investigate the transcriptomic effects on different pathways and reactions by including gene expression data into the proposed model.

GEMsplice (Angione, 2018) was used to merge gene expression data with the P. putida metabolic model. The main idea is to create a profile-specific metabolic model for each single gene expression profile. This is done by defining the constraints on fluxes in Eq. (1) as (2) Vmin φ(θ)≤v≤Vmax φ(θ)

where the function φ maps the expression level θ of each gene to a coefficient for the lower- and upper-bounds of the corresponding reactions, and is defined as (3) φ(θ)=[1+ γ|logθ|]sgn(θ−1)

where the sgn operator returns a vector of ±1 (signs of θ − 1). The constant γ sets the weight of the gene set expression level as an indicator of the rate of production of the associated enzyme (Angione, 2018). We ran our model with γ = 1 in order to ensure a logarithmic effect of the transcriptomic value on the flux bounds of the metabolic model.

We used the integrated model to investigate the relation between gene-expression data and rhamnolipids production in P. putida. We downloaded the expression data of P. putida from GEO (accession number: GSE28257). The dataset provides the expression levels θ of 5,547 genes for 40 samples of the P. putida wild type and 40 samples of P. putida Tn5 mutants. For each sample, a condition specific model was created by using Eq. (2) as constraint in the geometric FBA problem. We used maximum rhamnolipids production as the objective function in our engineered condition-specific models in order to maximize the production and export of rhamnolipids.

Elastic-net regression identifies key genes driving metabolic alterations

After running the condition-specific models, we compared the predicted flux rates (i.e., the FBA solution vector v) of the two groups (wild type and Tn5 mutant) to identify a set of differentially active reactions (DARs), that is, reactions with an adjusted p-value < 0.05. The identified reactions belong to disrupted metabolic pathways that carry a significantly different flux between the wild type samples and the Tn5 mutant samples when the rhamnolipids production is maximized. To further investigate those disrupted metabolic pathways and identify the genes contributing to the flux rates of the DARs in the two groups, we applied the variable selection regression method described below. The idea is to identify the genes that are highly predictive of the flux rate of each DAR when optimizing the rhamnolipids production rate.

Let t be the number of observations (samples) with p predictors (genes). Let y = (y1,…, yt)T be the response (the FBA solution vector) and X = (x1|…|xp) be the model matrix (gene expression matrix), where xj = (x1j,…, xtj)T, j = 1,…, p are the predictors. For any fixed non-negative λ1 and λ2, we use the elastic-net regularization criterion (Zou & Hastie, 2005), namely a linear combination of lasso and ridge regression penalties: (4) L(λ1, λ2, β) = ∥y−Xβ∥2+λ1∥β∥1+λ2∥β∥2

where β = (β1,…,βp) is the vector of coefficients to be estimated, ∥β∥1=∑j=1p|βj| and ∥β∥2=∑j=1pβj2. The elastic-net estimator β^ is the minimizer of Eq. (4): (5) β^=argminβ{L(λ1,λ2,β)}

Let α = λ2/(λ1 + λ2) and λ = λ1 + λ2; then solving β^ in Eq. (5) is equivalent to the optimization problem (6) β^=argminβ∥y−Xβ∥2+Pα,λ(β)

where Pα,λ(β) is the elastic-net penalty function defined as (7) Pα,λ(β)= λ[(1−α)∥β∥1+α∥β∥2]

In our analysis, the model matrix X was set equal to the normalized gene expression matrix, with t = 80 observations (expression profiles) and p = 5,547 predictors (genes). The response variable y was set equal to the vector of flux rates of the DAR to be analyzed (y is a vector with dimension t × 1). Hence, y is the vector with the flux rates of a given DAR resulting from running each of the t = 80 FBA condition-specific models. For each DAR, we set the regularization parameter α = 0.5 to achieve a balance between lasso and ridge regression. We used a 10-fold cross validation to identify the optimal λ. Simulations were carried out in R version 3.5.1 using the glmnet package 2.0-16 (Friedman, Hastie & Tibshirani, 2010).

Results

To implement the maximization of rhamnolipids production, we started from a genome-scale model of P. putida, iJP962 (Oberhardt et al., 2011). To enable the production of rhamnolipids, we engineered the iJP962 model by introducing the genes and reactions responsible for rhamnolipids biosynthesis from P. aeruginosa. Figure 1B shows the maximum production of biomass and maximum production and export of rhamnolipids. Using the reference condition (uptake of glucose at 10 mmol/gDW/h), our model predicted a production of 0.74 mmol/gDW/h of biomass, in agreement with the genome-scale model iJP962.

Figure 1B shows the rate of biomass and rhamnolipids production from the P. putida model simulated under different carbon sources (fructose, sucrose, glycerol, benzoate and myristic acid). We found that biomass synthesis and rhamnolipids production increased linearly with the rate of metabolite uptake. Our simulation-based predictive results are in keeping with our lab-based fermentation work previously carried out with Pseudomonas strains (Rahman et al., 2002, 2009, 2010; Joy, Rahman & Sharma, 2017; Parthipan et al., 2018). In addition, we also identified that myristic acid (C-14) provided optimal growth rate and rhamnolipids production compared to the other carbon sources in this study.

To pinpoint the key intermediates contributing to the formation of rhamnolipids, we assessed the pathways in the engineered model. Specifically, rhamnolipids utilize glucose-6-phosphate and acetyl-CoA (intermediates of central metabolism) to drive the biosynthetic pathway through two distinct routes: the rhamnose pathway and FA pathway (Fig. 1A). Glucose-6-phospate generated from degradation of glucose, fructose, sucrose and glycerol, when provided as a main carbon source, fed to the rhamnose pathway subsequently forming dTDP-rhamnose, a precursor of rhamnolipid. Consistent with the findings of Tiso et al. (2016), fluxes from the degradation of glucose, fructose and benzoate generated rhamnolipids via the rhamnose pathway. On the other hand, provision of myristic acid and benzoate entered FA degradation pathway, generating intermediates for RhlA to form HAA. Benzoate enters central metabolism via actyl-CoA and succinyl-CoA. This is in agreement with the report by Abdel-Mawgoud, Lépine & Déziel (2014). Subsequently, the RhlB formed rhamnolipids, which was then exported to the extracellular compartment (Fig. 1A). To determine the routes of rhamnolipids production by the P. putida model, we evaluated the flux distribution of the rhamnose and FA pathway. Simulation under all different carbon growth medium revealed that the flux through the rhamnose pathway was dominant in producing maximum amount of rhamnolipids compared to the FA pathway.

Rhamnolipids synthesis by the engineered model of P. putida

To determine the maximum rhamnolipids production by the engineered in silico model, we investigated several carbon sources and evaluated the metabolic network comprehensively (Fig. 1B). When the P. putida model was optimized for biomass and rhamnolipids production, the amount of rhamnolipids production increased with the uptake of each metabolite. Metabolism of myristic acid (C-14), followed by fructose and sucrose/glucose, provided the best condition for optimal rhamnolipids synthesis. As expected, rhamnolipids synthesis increased in a linear relationship with the increased uptake of various carbon sources (Fig. 1B). Our results also suggest that most of the rhamnolipids production derives from the rhamnose pathway rather than from the FA degradation pathway (Figs. 1A and 1C). Interestingly, when myristic acid was supplied as a carbon source, both pathways contributed to rhamnolipids production to a similar degree. Figure 1D shows rhamnolipids synthesis and biomass by the engineered model under each carbon source.

To determine whether a mixture of metabolites increased rhamnolipids production, we increased the uptake of mixed metabolites simultaneously. When glucose and glycerol or glucose and myristic acid were supplied as combinations of metabolites simultaneously, rhamnolipids production increased to 2.19 mmol/gDW/h and 4.50 mmol/gDW/h, respectively, compared to when each metabolite was supplied individually. In our previous study, a mineral salt medium used for growing biosurfactant producers was initially supplemented with two g/L glucose to initiate biomass production. This was followed by the addition of glycerol to test their influence on biosurfactant production. Pseudomonas aeruginosa DS10-129 produced a maximum of 1.77 g/L rhamnolipid with glycerol at 288 h (Rahman et al., 2002).

Comparison between P. putida and P. aeruginosa for production of rhamnolipids

Figure 1E shows the comparison between our model and the Pseudomonas aeruginosa PAO1 model (Oberhardt et al., 2008) in terms of rhamnolipids production under six different carbon sources (glucose, fructose, sucrose, glycerol, benzoate and myristic acid). The transport reactions for the export of rhamnolipids across the cell membrane were added to the P. aeruginosa model. In order to compare the two models under the same carbon sources, the transport reactions across inner and outer membranes for sucrose and benzoate were also included in the P. aeruginosa model (see Additional File 4 for the full list of reactions).

To analyze the different production rates of rhamnolipids, we investigated the alternative carbon sources separately by setting an uptake rate of 10 mmol/gDW/h for the carbon source under investigation, and zero uptake for the other sources. Both models show a high production rate of rhamnolipids when either glucose, fructose or sucrose is provided, consistent with previous results (Bahia et al., 2018). Glycerol provides enough nutrients for P. putida and P. aeruginosa for the production of rhamnolipids in accordance with Rahman et al. (2002) and Silva et al. (2010). When benzoate or myristic acid were provided as sole carbon source, the production rate of rhamnolipids was 1.967 mmol/gDW/h in P. putida and null in P. aeruginosa, which might be due to the unrelated genome codon index and codon adaptation index profiles of the two bacteria (Weinel et al., 2002). However, if we used the uptake of glucose at 10 mmol/gDW/h as a control growth condition (Oberhardt et al., 2011), the production rate of rhamnolipids was 1.818 mmol/gDW/h in both models.

Principal component analysis reveals biomarkers of rhamnolipids production in P. putida

Principal component analysis (PCA), a form of unsupervised machine learning, identifies data similarities from multidimensional biological datasets (Brunk et al., 2016). More specifically, PCA is a statistical technique that uses a multi-dimensional space to convert a set of correlated variables into linear uncorrelated latent variables called principal components. In our case, it is based on the singular value decomposition of the matrix of flux rates, and is therefore equivalent to finding the system of axes in the space of flux rates such that the covariance matrix is diagonal.

We investigated the individual reactions and identified the key components that drive change in the growth and rhamnolipids production in the engineered model. We applied PCA on our observed flux dataset generated under different growth media; glucose, fructose, sucrose, glycerol, benzoate and myristic acid. To characterize the unique features of individual reactions and variables in the observed flux datasets, we plotted the first two singular vectors of PCA (Figs. 2A and 2B). We found that the first two eigenvectors sum to 88.7% of the variance in the observed flux. These findings suggest that changes in reaction fluxes correlate with the availability of various carbon sources for growth and rhamnolipids production. Figure 2A shows the variable correlation plot of each variable and the contribution for the corresponding carbon source. We found that the first component correlates highly with the variables fructose and sucrose, while the second component correlates with the variables myristic acid and benzoate. Table 2A shows the detailed contribution of each carbon source on the principal components. These variations are driven by changes in the amount of carbon sources used for growth, indicating the network adaptation, particularly in the rate of core metabolic reactions.

Figure 2 Principal component analysis of flux rates of the engineered P. putida.

(A) Variables factor map. The distribution of each carbon sources used for growth correlates differently with the principal components. Fructose and sucrose correlate positively with the first component (Dim1), while the second component (Dim2) correlates highly with benzoate and myristic acid. (B) Individuals factor map. Key reactions of the central metabolism are drivers of growth and rhamnolipids production in the engineered P. putida model. Each component, RR08593 (ATP synthase) and IR10022 (cytochrome-c reductase) are distinguishable between the different conditions. The names of the top-30 reactions with the highest contributions have been reported in the factor map. (C) Correlation histogram. The distribution of each variable is shown in the diagonal panel representing the main carbon sources: glucose, fructose, sucrose, benzoate, glycerol and myristic acid. In the top panels, the absolute value of the correlation is shown with the result of the correlation test (p-value < 0.001). In the bottom panels, the bivariate scatter plots are displayed, with a fitted line. (D) Scree plot generated from eigenvalue versus component number. (E) Correlation matrix illustrating the correlation between each variable and the PCA latent dimensions. Blue color represents positive correlation, while the color intensity and size of the circles are proportional to the correlation coefficients. The reader is referred to Table 2 for individual reaction scores, and to the main text for further interpretation of the results.

Table 2 Contribution of variables and individual reactions on the principal components.

	PC1	PC2	PC3	PC4	PC5	
(A)	
Glucose	17.733	10.669	0.001	56.157	15.396	
Fructose	19.748	4.086	0.463	4.852	3.273	
Sucrose	19.167	1.467	4.929	34.892	29.010	
Benzoate	11.431	46.868	34.473	3.889	3.100	
Glycerol	18.208	10.804	0.978	0.101	48.304	
Myristic acid	13.713	26.106	59.157	0.108	0.917	
(B)	
RR08593	23.878	0.683	0.878	0.575	0.040	
RR08674	22.980	9.956	2.130	3.316	0.376	
EX_EC0001	17.651	7.972	2.223	0.260	0.772	
EX_EC0007	5.952	8.456	0.015	0.201	0.096	
IR10022	5.758	0.401	17.861	0.124	0.569	
RR04368	0.963	9.394	10.448	0.323	0.162	
Note:

(A) Fructose and sucrose are highly correlated with the first component (PC1), compared to benzoate or myristic acid. This variation is driven by the activity of core metabolic reactions for energy demand toward growth and rhamnolipid synthesis. (B) The individual reactions driving rhamnolipid synthesis scored highly on the first two principal components; these include ATP synthase (reaction id: RR08593), cytochrome-c reductase and succinate dehydrogenase (reaction id: IR10022 and RR04368).

Figure 2B shows the top-30 individual reactions, with the highest mean scores on the components, mapped on the first two principal components (for the full list of contributions see Additional File 5). ATP synthase (reaction id: RR08593) and cytochrome-c oxidase (reaction id: IR10022), together with the uptake of oxygen, H2O and H2O transport (reaction ids: EX_EC0001, EX_EC0007 and RR08674), scored highly with the first component, indicating the energy demand for growth and rhamnolipids synthesis (Table 2B). The utilization of cytochrome-c oxidase is a common feature of several proteobacteria (Osamura et al., 2017); it is involved in the production of ATP via the respiratory electron transport chain and contributes to the production of the necessary enzymes subsequently used for ATP production by the ATP synthases (Tremblay & Déziel, 2010). One of the reactions that scored highly with the second component is succinate dehydrogenase (reaction id: RR04368). This reaction is involved both in the tricarboxylic acid cycle (TCA) and in respiration via the electron transport chain linked to rhamnolipid production (Wittgens et al., 2011). Figure 2C reports the correlation matrix of the six variables under investigation (the distribution of each variable, the absolute value of the correlation, the result of the correlation test and the bivariate scatterplots with a fitted line). The plot shows that the results of this preliminary analysis are in accordance with the results reported in the PCA variables factor map (Fig. 2A).

In order to analyze the quality of our PCA analysis, we report the scree plot (Fig. 2D) and the cos2 correlation map (Fig. 2E). The first two components retain 88.7% of the information (variances) contained in the data, which allows us to focus only them for the statistical analysis of the model. Moreover, the correlation plot of cos2 (Fig. 2E) indicates a good representation of the variables on the first two principal components. This also explains the position of the six variables in Fig. 2A (they are close to the circumference of the correlation circle).

In conclusion, the PCA analysis shows that our results are in agreement with those obtained by Wittgens et al. (2011) and Tiso et al. (2016). Indeed, the high cos2 value of glucose and fructose shows that they both play a key role in the metabolic pathway of rhamnolipids synthesis chosen as FBA objective. Hence, fluxes from the degradation of glucose and fructose generate rhamnolipids via the rhamnose pathway.

Regression analysis identifies disrupted pathways and genes

We integrated gene expression profiles into the proposed metabolic model of P. putida to investigate disrupted metabolic reactions and pathways using GEMsplice (Angione, 2018). We compared 40 samples of the P. putida wild type with 40 samples of P. putida Tn5 mutants. We simulated the growth medium using the single carbon source that allowed the highest production of rhamnolipids, that is, myristc acid (Fig. 1C). Hence, we set the uptake of myristic acid equal to 10 mmol/gDW/h while the uptake of the other sources was set equal to zero. Table 3 reports the list of the pathways associated with the top-5 DARs, that is, reactions with an adjusted p-value < 0.05 (for the full list of 15 DARs see Additional File 6).

Table 3 Top-5 differentially active reactions (DARs) in wild type and Tn5 mutant P. putida samples.

Top-5 Differentially active reactions	Pathways	p-value	
L-Alanine:3-oxopropanoate aminotransferase	Purine metabolism	0.0042	
3-Aminopropanoate:2-oxoglutarate aminotransferase	Fatty acid biosynthesis	0.0043	
4a-hydroxytetrahydrobiopterin dehydratase	Benzoate degradation via hydroxylation	0.0066	
NADH:6,7-dihydropteridine oxidoreductase	Pyrimidine metabolism	0.0341	
L-Phenylalanine,tetrahydrobiopterin:oxygen oxidoreductase(4-hydroxylating)	Folate biosynthesis	0.0341	
Note:

The first column reports the list of the top-5 DARs (reactions with adjusted p-value < 0.05). The second column report the pathways associated to each disrupted reaction. The adjusted p-value is reported in the third column.

Purine metabolism and FA biosynthesis are the two pathways associated with the top-2 DARs with adjusted p-value of 0.0042 and 0.0043, respectively. Both pathways play a key role in the rhamnolipids production (Rehm, Mitsky & Steinbüchel, 2001) and bacterial membrane biogenesis (Zhang & Rock, 2012). The benzoate degradation via hydroxylation pathway (adjusted p-value = 0.0066) has also been previously linked to the rhamnolipids pathways (Procópio et al., 2012). Indeed, the genes encoding enzymes involved in the rhamnolipids productions also encode enzymes for the benzoate degradation via hydroxylation pathway. It is noteworthy that the identified DARs reflect the disruption of metabolic pathways from the interaction between gene expression profiles (integrated through GEMsplice) and metabolic networks (represented by the P. putida metabolic model). As a consequence, these results are complementary to the outcomes obtained through metabolic network analysis alone, which does not take into account specific transcriptomic profiles.

Regression analysis (elastic-net, see Methods) was then applied to identify the key genes contributing to the flux rate of the DARs. Figure 3 shows the distribution of the top-10 genes (genes with the highest |β|) c the most disrupted DAR (in the purine metabolism pathway) for the wild type and Tn5 mutant samples. By analyzing these distributions it is possible to characterize the metabolic diversity of the different samples and predict their behavior under different conditions. For example, the different distribution of the gene PP2431 might reveal a different cellular adaptation (Fernández et al., 2013). Moreover, the gene PP4355 has been identified as a gene involved in the encoding process of diverse flagellar components in Tn5 mutants samples, which might explain the different distributions in the two types of samples (Sharma et al., 2014). We stress that the procedure proposed here is a single case study, and it can be adapted and extended to identify or compare any two different types of P. putida samples.

Figure 3 Top-10 genes contributing to the flux of the most disrupted metabolic reaction (part of the purine metabolism pathway).

The boxplots report the distribution of the 10 genes with the highest |β| resulting from the elastic-net regression analysis in both wild type and Tn5 mutant P. putida samples.

Discussion

The growing demand for biosurfactants requires rapid, efficient and innovative approaches for its synthesis, including the use of microorganisms. However, native bacterial cells are very inefficient at maximizing the production of industrially-relevant products. Bioengineering of such cells can improve the yield, but the number of potential metabolic and genetic interventions is enormous in practice (Kell, 2012). At the same time, the emergence of in silico modelling enables us to metabolic engineer microbial networks in silico, and to predict their efficiency in a variety of growth conditions.

Machine learning tools coupled with computational modelling of metabolism can rapidly identify ways of increasing the productivity of these cells toward maximum production of biosurfactants while maximizing the growth rate of the cultures. In this study we genetically engineered P. putida KT2440, offcially classified as a “generally recognized as safe” strain and used in the production of diverse natural products, including rhamnolipids (Loeschcke & Thies, 2015). In particular, P. putida was observed to have resistance to higher rhamnolipid concentrations (90 g/L) in the production medium when compared to other microbial hosts of industrial importance like E. coli, B. subtilis and C. glutamicum (Wittgens et al., 2011).

Recombinant rhamnolipid production has many industrial advantages, including the opportunity to use non-pathogenic production strains and the ability to produce rhamnolipids independent of the complex quorum sensing regulation. Non-pathogenic bacterial strains have been genetically engineered to express P. aeruginosa rhl-genes for the heterologous rhamnolipid production (Beuker et al., 2016). Ochsner et al. (1995) studied rhamnolipid synthesis by recombinant P. fluorescens, P. putida, P. oleovorans and E. coli with the rhlAB operon from P. aeruginosa and observed rhamnolipid production by P. fluorescens (0.25 g/L) and P. putida (0.6 g/L). But no rhamnolipids were produced by recombinant E. coli and P. oleovorans, despite the detection of an active rhamnosyltransferase. Recombinant E. coli strains were also used by Wang et al. (2007) and Cabrera-Valladares et al. (2006) for heterologous expression of P. aeruginosa rhlAB genes. Cha et al. (2008) and Cabrera-Valladares et al. (2006) reported rhamnolipid production by a recombinant P. putida (7.3 g/L) and recombinant E. coli HB101 (52 mg/L) with soybean oil and oleic acid as substrates, respectively. As the production of high yields of rhamnolipids is dependent upon precursors provided by the metabolic flux within the bacterium, it is unlikely that simply implanting the necessary genes in a bacterium will be sufficient to make that organism produce rhamnolipids in higher concentrations (Marchant & Banat, 2012). Genetic alterations can however be an important part of organism selection for fermentation processes, and computational tools can help finding the best experimental setting to maximize their production.

In this study, we have taken a genome-scale approach to investigate the metabolic potential of P. putida to produce rhamnolipids by optimizing multiple cellular functions. Figure 1B shows the rate of biomass and rhamnolipids production by P. putida model simulated under different carbon sources: glucose, fructose, sucrose, glycerol, benzoate and myristic acid. Biomass synthesis and rhamnolipids production increased linearly with the rate of metabolite uptake, and myristic acid (C-14) supported optimal growth rate and rhamnolipids production compared to the other carbon sources. When the P. putida model was optimized for biomass and rhamnolipids production, the amount of rhamnolipids production increased with the uptake of each metabolite. Quorum sensing, namely the mechanism by which bacteria engage in cell-to-cell signaling communication using diffusible molecules based on a critical cell density, might be one of the reasons why rhamnolipid synthesis is associated with exponential stage of the biomass (Dusane et al., 2010).

These outcomes support previous studies involving rhamnolipid production on sugars and sugar-containing wastes. Sugar-containing wastes are gaining prominence due to their lower cost when compared to oil- or glycerol-containing wastes despite the lower rhamnolipid yields (Henkel et al., 2012). Agro-industrial wastes are rich in carbohydrates and lipids and hence can be used as a carbon source for microbial growth and rhamnolipid synthesis (Gudiña et al., 2015). Among them, molasses has a high sucrose concentration in the range of 50–55% by weight. Raza et al. (2007) obtained a maximum of 1.45 g/L rhamnolipid yield after 96 h of incubation with P. aeruginosa EBN-8 mutant on 2% blackstrap molasses. Similarly, Onbasli & Aslim (2009) used 5% sugar beet molasses and obtained a maximum rhamnolipid yield after a 12 h incubation with P. luteola B17 and P. putida. Li et al. (2011) and Gudiña et al. (2015) observed the highest biosurfactant production yield of 2.6 g/L and 3.2 g/L by P. aeruginosa using molasses distillery wastewater and a culture medium containing corn steep liquor and molasses, respectively.

We observed that the metabolism of myristic acid provided the best condition for optimal rhamnolipids synthesis, followed by fructose and sucrose/glucose. Plant oils are a rich source of myristic acid and these long chain FAs have been successfully used as carbon source for rhamnolipid biosynthesis. For instance, Radzuan, Banat & Winterburn (2017) showed that P. aeruginosa PAO1 can grow and produce 0.43 g/L of rhamnolipids using palm FA distillate under batch fermentation. Cha et al. (2008) studied the growth of P. aeruginosa EMS1 and P. putida 1,067 in mineral salt medium with 2% soybean oil as the sole carbon source. They detected rhamnolipid productions of about 5.18 g/L and 6.97 g/L, respectively. This shows that P. putida 1,067 is more efficient than P. aeruginosa EMSI in using plant oils as carbon source. Vegetable oils are more efficient in inducing rhamnolipid production when compared to the hydrophilic substrates like glucose, fructose and sucrose; this may be due to their water-soluble nature that facilitates the ease of uptake. However, vegetable oils are hydrophobic, and this stimulates the bacterial rhamnolipid production to increase their solubility (Cha et al., 2008).

When we investigated the metabolic reactions and pathways that are disrupted by integrating the gene expression profiles into the proposed metabolic model of P. Putida, the top two pathways observed were purine metabolism and FA biosynthesis, followed by benzoate degradation, pyrimidine metabolism, folate biosynthesis and porphyrin and chlorophyll metabolism. These results highlight the essential role of nucleic acid metabolic pathways in rhamnolipid biosynthesis. This might be due to the fact that under exponential growth conditions bacterial replications leads to the activation of purine and pyrimidine pathways. Moreover, porphyrin and chlorophyll metabolism play an important role in the biosynthesis of tetrapyrroles like hemes, chlorophylls and cobalamin. They serve as prosthetic group of many proteins involved in fundamental biological processes like respiration, metabolism and transport of oxygen. Further, heme acts as essential cofactor for enzymes such as catalases, peroxidases and cytochromes.

Nikel & De Lorenzo (in press) have recently published an updated genome annotation of P. putida KT2440, which includes novel catabolic pathways for 32 carbon sources, 28 nitrogen sources, 29 phosphorus sources and 3 carbon and nitrogen sources. This unique metabolic architecture of P. putida will be harnessed for future studies. Furthermore, while FBA only allows for one objective function (usually the growth rate), multi-target optimization algorithms have been developed and applied to genome-scale metabolic models of microorganisms to optimize multiple cellular functions (Costanza et al., 2012; Angione & Lió, 2015). We specifically envisage the use of machine learning coupled with multi-level optimization for industrial biotechnology. For instance, one can engineer a microorganism to maximize the export of selected chemicals out of the cellular membrane, while ensuring biomass production and simultaneously minimizing byproduct formation. Taken together, our findings clearly show the potential use of engineered strains coupled with metabolic modelling and machine learning tools for rhamnolipids production.

Conclusion

We engineered a genome-scale model of P. putida for optimization of rhamnolipids production as a high-end secondary metabolite. Our in silico model was engineered to produce rhamnolipids by utilizing two key enzymes: RhlA and RhlB. All corresponding biochemical reactions for rhamnolipids biosynthesis were added from the P. aeruginosa model (Oberhardt et al., 2008); where appropriate, KEGG, MetaCyc and BRENDA were used to add new reactions for the rhamnose pathway. Our engineered in silico model was designed to synthesize and export rhamnolipids; the transport mechanisms for rhamnolipids export were modelled as a reaction step that carried out fluxes from the intracellular to the extracellular compartment across the cell membrane. The engineered model was manually curated and geometric FBA was used to reproduce the flux of 0.74 mmol/gDW/h of biomass, consistent with the iJP962 model (Oberhardt et al., 2011).

A further statistical analysis based on PCA was performed to further elucidate the metabolic behavior, and to identify roles of individual nutrients and reactions in shaping the response of the engineered cell. Finally, transcriptomic data was integrated into our model, which allowed building condition-specific models of P. putida to exploit and predict the metabolic and genetic engineering steps needed for maximizing rhamnolipids production. These models were investigated with elastic-net regression with the aim of identifying latent pathways and genes correlated with enhanced production of rhamnolipids in P. putida.

When experimental data on the engineered organism become available, we envisage three directions for extension of the model. (i) Multi-step optimization algorithms can be used to maximize the growth rate and rhamnolipids synthesis, and simultaneously minimize byproduct formation in a multi-target fashion (Angione, Pratanwanich & Lió, 2015). The proposed in silico design of P. putida can then be assessed using advanced sensitivity techniques, robustness and control analysis. (ii) If more than one omic-level information is available, methods from network theory (Angione, Conway & Lió, 2016) can be adapted to give insights into the model and predict the behavior of the microorganism in untested conditions. (iii) Alterations to the regulatory genes RhlI and RhlR could influence overall yield. In addition to this, the RhlC gene, which codes for a rhamnosyl transferase responsible for mono- to di-rhamnolipid conversion, could be regulated (partial knockdown or overexpression) to ensure the production of a specific type of rhamnolipid.

The ability to adapt to such conditions across multiple omic levels can for instance be assessed by evaluating the changes in the proteins of the outer membrane, key players in the adaptation of Pseudomonas to environmental perturbations and in the production of rhamnolipids (Wilhelm et al., 2007; Bouffartigues et al., 2011). Taken together, our findings give strong basis for metabolic engineering of P. putida for rhamnolipids production and provide a framework and a working model for further studies involving optimization of biosurfactant production.

Supplemental Information

Supplemental Information 1 List of reactions added to the model.

Click here for additional data file.

Supplemental Information 2 P. Putida engineered metabolic model in Matlab format.

Click here for additional data file.

Supplemental Information 3 Matlab script to generate the flux distributions.

Click here for additional data file.

Supplemental Information 4 Script for adding missing reactions in P. Aeruginosa.

Click here for additional data file.

Supplemental Information 5 PCA contributions of reactions and variables.

Click here for additional data file.

Supplemental Information 6 Differentially Active Reactions (DARs) and their pathways.

Click here for additional data file.

List of abbreviations

FBA flux balance analysis

QS quorum sensing

FA fatty acids

PCA principal component analysis

DARs differentially active reactions

Additional Information and Declarations

Competing Interests

Author Contributions

Data Availability

The authors declare that they have no competing interests.

Annalisa Occhipinti performed the experiments, analyzed the data, prepared figures and/or tables, authored or reviewed drafts of the paper, approved the final draft, built the models and wrote the code.

Filmon Eyassu performed the experiments, analyzed the data, prepared figures and/or tables, authored or reviewed drafts of the paper, approved the final draft, built the models and wrote the code.

Thahira J. Rahman authored or reviewed drafts of the paper, approved the final draft.

Pattanathu K. S. M. Rahman authored or reviewed drafts of the paper, approved the final draft, contributed to select the substrates for the proposed models using laboratory experimental results.

Claudio Angione conceived and designed the experiments, coordinated the study,analyzed the data, authored or reviewed drafts of the paper, approved the final draft, built the models and wrote the code.

The following information was supplied regarding data availability:

The raw data are provided in the Supplemental Files.

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
