# Peer review of "In silico engineering of Pseudomonas metabolism reveals new biomarkers for increased biosurfactant production"

_PeerJ, doi:10.7717/peerj.6046_

## Round 0.1 · original submission · Major Revisions

Reviewer 2 raises several issues regarding the claims of novelty and the completeness of the documentation of the modeling methods used. The manuscript states: "To the best of our knowledge, this is the first study that uses in silico modelling of P. putida for optimizing rhamnolipids synthesis." However, the cited work by Wittgens et al. (2011) clearly uses flux balance analysis to perform in silico analyses of P. putida rhamnolipid production (see quote below).

"Having the chassis, the design of a metabolic network with high capacity for rhamnolipid synthesis in P. putida, using flux balance analysis with the rate of rhamnolipid production as linear programming objective, was in focus. The constraints of the metabolic network were, besides its structure, the substrate uptake rate, the rate of biomass formation, and the energetic demand for cell maintenance. The theoretically achievable yields of rhamnolipids on industrial relevant substrates (glucose, sucrose, glycerol, and fatty acids (here as example octanoate)) were estimated. The computational results indicate that cell growth should be minimized to achieve high rhamnolipid yields..." (From Wittgens et al. (2011)).

Given the prior work, it is important to 1) clarify how this work differs from or builds on that research, and 2) clearly document the modeling methods utilized.

Please give care to address the comments of both reviewers in your revisions, but especially the concerns raised by Reviewer 2. Be sure to clarify what software tools were used and how the provided supplemental code can be executed by reviewers/readers. We look forward to receiving your revised manuscript.

Reviewer 1 ·

Basic reporting

Figure 2A should be edited in order to improve readability, (probably name of reactions can be hidden since they are listed separately).

Experimental design

The work here reported is well described and the method used are innovative, however I would suggest the following point to address in order to improve the quality of the manuscript:

- the new production rate of rhamnolipids should be compared not only to that predicted by the previous version of the model but also to the one obtained through P. aerugionosa model (from which reactions have been taken).

- a detailed list of genes and reactions identifiers added to the model is missing, such data should be provided alongside a functional description of genes added.

- line 75, no reason for such a transfer of genes is reported, could you provide any example of production of rhamnolipids in other strains by mean of the named genes?

Validity of the findings

The manuscript would benefit of a comparison between P. putida and P. aeruginosa performance in rhamnolipids production and would eventually support the suggested P. putida engineering.

Reviewer 2 ·

Basic reporting

The English is professional, but be careful to include all necessary articles. There are a few missing 'the's scattered through the text.

Experimental design

The claims of novelty in the introduction are not justified, as the authors cite several other studies that offer more rigorous assessments (both in silico and in vitro) of optimized rhamnolipid production than what they here show (particularly Wittgens et al. 2011., but also Tiso et al. 2016. and many more papers optimizing rhamnolipid production in P. aeruginosa and stutzeri). I do not feel that the study fills an identified knowledge gap.

There is also far too little detail in the methods to assess how the modeling was performed. The provided supplemental code is incredibly bare and inadequate, and the 'flux_balance' function as presented is not, to my knowledge, a part of the Cobra Toolbox which they claim to use in this analysis. I therefore can't run the minimal code they present to even figure out what is going on, though I can optimize their provided model using other Cobra Toolbox functions. There are also many complex implementations of multi-objective optimizations that they here seem to bypass in favor of the multi-level approach used by Angione et al. 2015, and the rigor of the former approaches in addition to the lack of methods detail is concerning. Furthermore, the lack of description regarding the 'flux distribution' they are analysing with PCA makes me think that they have not performed any flux sampling, flux variability analysis, or other approaches that would ensure that they have obtained a fixed solution versus one of the many possible flux distributions which may fulfill their constraints. This issue must be addressed for any findings to be valid.

Validity of the findings

As the addition of the rhamnolipid pathway to the P. putida model is a straightforward and relatively easy process, the main scientific advances claimed by this study seem to be 1) the multi-level optimization balancing growth and rhamnolipid production (which I cannot properly assess given the above methods issues, and there are easier ways to perform this by tracing the pareto front of the two objectives) and 2) the PCA assessment of the 'flux datasets.' I have trouble following the PCA analysis, particularly what 2B is showing and how it relates to the PCA. They furthermore do not even address Figures 2D-E in the text. They also provide little discussion or contextualization of the PCA results, likely because the reactions identified are not particularly interesting or relevant to rhamnolipid production.

Additional comments

Calling PCA machine learning is technically accurate, but quite a stretch in common practice given the much more advanced machine learning being implemented in biology today.

---

## Round 0.2 · accepted · Accept

Thank you for your significant effort in responding to the reviewers' and editor's comments. The paper has been improved and I concur with Reviewer 1 that it is now ready for publication.

# Reviewer 1 ·

Basic reporting

no comment

Experimental design

no comment

Validity of the findings

no comment

Additional comments

The authors extensively improved the quality of the manuscript by including litterature references as a background and a clear work rational in the introduction.
They performed the suggested comparison and edited the figures in order to improve readability.